# The Effects of ADAS on Driving Behavior: A Case Study

**DOI:** 10.3390/s23041758

**Published:** 2023-02-04

**Authors:** Gaetano Bosurgi, Orazio Pellegrino, Alessia Ruggeri, Giuseppe Sollazzo

**Affiliations:** Department of Engineering, University of Messina, 98166 Messina, Italy

**Keywords:** vehicle–infrastructure cooperation, connected driving, intelligent road vehicles, driving behavior, road safety

## Abstract

The presence of numerous sensors inside modern vehicles leads to the development of new driving assistance tools, the real usefulness of which depends, however, on the environmental context. This study proposes a procedure capable of quantifying the effectiveness of some warnings produced by an On-Board Unit (OBU) inside the vehicle in a specific environmental context, even if limited only to the considered road. The experimentation was carried out by means of a driving simulator with a sample of young users with sufficiently homogeneous characteristics. The collected data were treated by ANOVA to highlight any differentiation between a traditional driving condition, without any instrumental support, and another involving the OBU was present. The results showed that only in relation to the investigated road, the OBU ensured the advantage of sending information of interest to the driver without invalidating their performance in terms of longitudinal and transverse acceleration, speeding, and steering angle. This research could be of interest to the infrastructure managers who, in case of inappropriate use of a road, could intensify active and passive safety devices for users’ safety.

## 1. Introduction

Traditionally, the road system has four components: human, vehicle, road, and environment. The human component is difficult to characterize objectively due to the many involved variables, such as age, driving ability, and propensity to accidents. Moreover, car drivers are non-professional and a driving license does not ensure safe driving.

Especially in recent years, the automotive industry has developed in-car devices for entertainment and safety purposes by actively or passively supporting the driver during their activity. However, these devices can affect the driver’s cognitive responses while driving.

Indeed, infotainment and active-safety devices (i.e., those requiring control and action by the user) have a common feature: both, unfortunately, absorb resources from the driver, reducing those available for sudden needs while driving [1,2,3,4]. The potential inattention determined by infotainment devices caused a good part of the road accidents [5] due to a series of issues, such as delayed perception, reduction of the time-to-collision, abrupt slowdowns, decrease in headway times, and invasions of opposite lanes [6,7,8,9,10]. 

Dividing the information flow into different perception channels, such as visual, auditory, and tactile can help avoid overloading one of them. However, vision represents the main sense through which all the surveyed data are acquired by drivers [11,12,13,14,15,16,17].

In the past, in particular contexts such as air traffic control or nuclear power plants, the role of the human being has been studied more carefully because, when subjected to heavy stress in hostile environments, they may have reactions leading to catastrophic events. This kind of stress has been called workload and its meaning has been extended to other challenging environments, such as roads. The definition provided by O’Donnel and Eggemeier [18], according to which the workload is the portion of the operator’s capacity required to perform a particular action, remains valid. From here on, due to an ever-wider application and the need to define a quantification methodology, numerous measurement methods have become widespread. These must positively respond to several needs, including:-Sensitivity even to small changes in the reference scenario;-Diagnostic to identify the cause of possible performance decay;-Ability to recognize interdependencies between variables;-Reliability and reproducibility of the obtained results.

A generally accepted classification by the scientific community divides all the measurements into three large groups: subjective, performance, and physiological.

Subjective measurements generally refer to questionnaires that are administered to users before or, more frequently, after their task. They have some advantages, besides the low cost, as they can deepen some particular dimensions of the user’s character, hidden from instrumental measurements. It should be noted that the workload depends not only on the external context in which the task takes place, but also on how this is perceived and performed by the user. Among their disadvantages, there is certainly an extreme subjectivity; moreover, providing the survey at the end of (and not during) the test does not help; then, some critical issues can be neglected or scaled down within an overall judgment [19].

The performance measures are obtained from vehicle telemetry and are the easiest to interpret. Any difference from design standards assumptions (e.g., axis line, operating speed, longitudinal and transverse acceleration, steering control) can indicate that the driver’s activity is deviating from the ideal conditions [20,21,22,23,24]. This type of measure may not provide a complete judgment since the task could be completed by two different users with different levels of stress, depending on their driving skills.

The third category of measures concerns the physiological ones, which always have the advantage of relying on an instrumental quantification but are more complex to interpret because they concern the sphere of driver’s psychophysiology. Among the most used variables [16,22,25], there are those related to visual behavior (fixations, saccades, pupil diameter), emotion control (heart rate, dermal conductivity), and muscle stress (electromyography). Another advantage of these measures is the continuity of measures while driving; thus, it is simple to report them to other aspects related to the investigated scenario.

Another difficulty regarding latency, i.e., the delay with which an effect occurs with respect to its triggering cause, is that it can vary according to the user and the problem type.

Driving can be mentally tiring, and the possible consequences of this psycho-physical state could be catastrophic. The technological evolution of instruments and devices inside the vehicle and on the roadside, in terms of the number of information transferred to the driver, can represent a further service for users, but also an unsustainable overload, especially in some particularly different conditions (weather, visibility, traffic, etc.). Then, these devices must be tested in advance, and possibly in a simulated environment, to ensure greater safety on critical roads for accident propensity. Then, the results of these experiments must be related to the previously reported workload measurements.

### Research Gap

The automotive industry is bringing increasingly sophisticated in-vehicle instrumentation to the market, the impact of which on drivers needs to be carefully evaluated.

While infrastructure managers aim to fully digitalize the roads under their jurisdiction [26,27], which may exacerbate any critical issues, this paper proposes a methodology that quantifies both performance and subjective measures to determine the drivers’ workload induced by these assistive instruments.

This methodology does not verify the effectiveness of driver assistance systems in absolute terms, but rather evaluates it on a case-by-case basis for a specific class of users and in a given driving context.

The problem is complex as it depends on many variables, such as gender, age, culture, driving experience, driving tendency, road geometry, and visual conditions. Therefore, in order to restrict the validity of the results to a specific context, some of these variables must be fixed for each scenario.

Despite the small sample size, the main purpose of the research can still be achieved as it concerns the proposal of a working methodology rather than the identification of general results that can be applied to every context.

In this regard, if a specific road section must be investigated, it would be advisable to build a larger sample of users, identifying the most fragile class of users on that road (it could be represented by the elderly, for example). This characteristic should be ascertained through a specific accident analysis. Regarding the device to be tested, its modifications could be considered if the results are not satisfactory. If the critical issues do not concern the device but the infrastructure, then the road manager will have to make the necessary corrections to the road context or insert road traffic restrictions [28].

In particular, the aim is to verify the effectiveness of these instruments and, if so, their validity in two scenarios of different complexity [29,30,31]., it will be necessary to test these scenarios, even in the absence of such instrumentation (Figure 1).

The details of the experimental phase are reported in the Section 2. In particular, Section 2.1 “General Description of the Experiment” describes the object of the experiment and the methods for its implementation. In this section, the four experimental conditions, characterized by different levels of complexity, will be described and statistically analyzed.

Then, in Section 2.2, the main characteristics of the driving simulator are reported, while in the following one (Section 2.3 “The Users’ Sample”) details on the sample of drivers are provided. Section 2.4, the “NASA TLX Questionnaire”, presents details on the first comparison tool. To evaluate the workload, at least two of the subjective, physiological, and performance methodologies should be used to validate the final considerations. In this case, the NASA TLX questionnaire, well known in literature, will be used together with performance measures and examined through the ANOVA test (Section 2.5).

## 2. Materials and Methods

### 2.1. General Description of the Experiment

The trial was conducted through a driving simulator located at the Digital Laboratory of Road Safety (DiLaRS) of the University of Messina.

The use of a simulated environment, as opposed to a real one, offers several advantages, including:-Repeatability and homogeneity of the environmental conditions (light, weather and traffic);-Users’ safety (including other vehicles);-The possibility of representing situations with details that can hardly be found in real contexts;-Complete and accurate vehicle telemetry;-The ability to equip the driver with non-invasive biometric sensors without compromising their safety and avoiding the risk of sanctions.

The simulated environment was based on an existing rural road, located within the municipality of Messina, Italy, known as SS113. This road connects the localities of “Colli San Rizzo” and “Locanda” (Figure 2) and is characterized by a winding alignment, not fully in compliance with modern road standards. Specifically, the curves are composed only of circular arcs without transition elements and have inconsistent radii. The road cross section, which consists of two lanes and two small shoulders, with a total width of about 7 m. visibility, in isolated vehicle conditions, is hindered by irregular horizontal and vertical geometry. Road signs and markings constitute an important source of information in a critical context, which can be further exacerbated by adverse weather conditions or traffic.

As stated in the introduction, the objective of this study is to evaluate the effectiveness of an on-board assistance instrumentation device, referred to as an on-board unit (OBU), under at least two different conditions. The first one is characterized by isolated driving (without traffic), in which the only source of stress is the alignment of the road. The second condition involves the addition of cycling traffic along the same lane, thus requiring the driver to deal with small groups of three or four cyclists.

The OBU (Figure 3) was programmed using Python language and simulates an assistance device through the provision of warnings for imminent dangers or road use violations. The OBU screen may also serve as a terminal for communication between vehicle and infrastructure (so-called V2I) or between vehicles (V2V), assuming that the digitization of the infrastructure has been completed.

The OBU screen remains empty if the user adheres to the road use rules, corrects potentially risky behaviors, or if there are no irregular geometric elements on the road that may pose a danger [30].

As shown in Figure 3a, it is possible to identify four quadrants in which feedback related to inappropriate and/or dangerous driving behavior is displayed in dials 1, 2, and 3. In quadrant 4, on the other hand, informative feedback on some critical aspects of the road layout is displayed. The indications displayed on the OBU were deemed to be adequate and clear for the users, even though the images displayed on the figure showing the simulator screen may not perfectly represent the actual visibility conditions. At the end of the experimentation, users were asked to fill out an additional questionnaire to identify any operational and perception difficulties encountered during the experimentation. No issues were reported regarding the comprehensibility of the information provided by the OBU.

Obviously, the trials performed with the OBU must also be evaluated in its absence to estimate its effective benefit.

Therefore, four different driving conditions can be identified:Control condition, no traffic.;Control condition, with traffic (cyclists);Smart condition (the OBU is active), no traffic;Smart condition (the OBU is active), with traffic (cyclists).

The OBU screen was only visible in two of the four conditions (i.e., conditions 3 and 4). In the conditions where the screen does not appear (conditions 1 and 2), the driver must rely on information outside the cockpit (Figure 4).

To simplify the environmental context and to produce more robust statistical results, optimal weather and light conditions (i.e., sunny weather and full daylight) were considered. No other distracting elements were added (traffic in the opposite direction, unevenness of the pavement, obstruction to visibility, etc.).

The vehicle used for the test is a very popular model (Citroen C3) commonly found on Italian roads.

### 2.2. The Driving Simulator of the University of Messina

The driving simulator is called SimEASY^®^, produced by AVSimulation, and is located in the DiLaRS of the University of Messina (Figure 3). This simulator has the following main features:-Three 29-inch full HD screens;-Steering wheel featuring a force feedback sensor to simulate the rolling motion of wheels and bumps;-Accelerator, brake, and clutch pedals, with manual or paddle shift on the steering wheel;-Sound effects played through different speakers and subwoofers;-Software called Scaner Studio^®^, used to design roads, generate environmental context and perform tests;-Data collected at a frequency of 10 Hz.

### 2.3. The Users’ Sample

The sample of users was selected from the student population to ensure a high degree of homogeneity in characteristics. In particular, only the trials of 14 users (4 women and 10 men) were considered for the following analysis, with an average age of 27 years, an average driving license age of 8 years, no significant visual pathology (myopia less than 1 diopter), no history of accidents suffered or caused, and no prior experience with driving simulators.

The initial group of participants was larger, but some were excluded due to significant visual impairments (excessive myopia) or reported instances of nausea during simulated driving.

To ensure familiarity with the driving controls, a training path of approximately 15 min was prepared prior to the actual testing.

All the trials were conducted according to the American Psychological Association Code of Ethics and after obtaining informed consent from each participant.

### 2.4. NASA TLX Questionnaire

The workload was determined through measures of performance of the human-vehicle system, as described in the following sections. To ensure consistency between the two classes of measures, a subjective survey was also conducted by administering the NASA TLX questionnaire.

Specifically, the users completed the questionnaire at the end of the first two driving conditions, called “Control” (without assistance by the OBU), and at the end of the last two conditions, called “Smart” (characterized by the presence of the OBU).

As is widely known, the NASA TLX questionnaire has a multi-dimensional feature, consisting of six subgroups representing variables that are somewhat independent, such as mental, physical, and temporal demands, and frustration, effort, and performance. The multidimensional scale provides an overall workload score on a 100-point scale, based on the weighted average of the six subscales.

At the end of the survey, the overall workload (OW) was determined, representing the total workload to which each user was subjected.

### 2.5. One-Way ANOVAs

The data set regarding the users’ activity was then analyzed using one-way ANOVAs to establish (1) the impact of the presence or absence of smart alerts, (2) the impact of the presence or absence of cyclists, and (3) which of these factors is the most critical.

The independent variables considered were:-Driving condition (2 levels: Control and Smart);-Traffic (2 levels: no cyclists, with cyclists).

The response variable (or dependent variable, DV) was alternatively represented by the following measures:-Longitudinal Acceleration (Acc X): it concerns, above all, the decelerations that can represent an indicator of a non-homogeneous driving behavior;-Lateral Acceleration (Acc Y): excessive values can be determined by the absence of transition curves that, when present, ensure a more gradual trend. As is known, its derivative with respect to time is the jerk, which is a symptom of users’ discomfort;-Road Gap: the distance from the centerline. The vehicle is unable to maintain the centerline of its lane accurately due to the absence of transition curves, which leads to a deviation of the trajectory from the centerline, which could represent a significant indicator of a user’s safety;-Speed of the vehicle: too low speed could be a symptom of difficulty in interpreting the road geometry;-Steer speed: this variable could indicate sharp maneuvers by drivers to correct non-optimal trajectories. In the presence of cyclists, they can provide an idea of how the passing maneuver was performed.-Overall Workload (OW): The NASA TLX questionnaire was administered to each user at the end of the guide in control condition and at the end of the guide in smart conditions.

The reliability of the results depends on the satisfaction of the assumptions of the ANOVA analysis. These assumptions include:-The dependent variable must be measured at the continuous level.-The independent variable should consist of at least two related groups that indicate that the same subjects are present in both groups.-The observations are independent, without a relationship between the observations in each group or between the groups themselves.-The absence of significant outliers.-Tests for normality by means of residuals.-Checking that the sphericity, i.e., the variances of the differences between all combinations of related groups, were equal. When these conditions are violated, the Mauchly tests for sphericity can be performed, adjusting the analysis by a correction criterion as the Greenhouse–Geisser method.

The following pairs of null or alternative hypotheses must be verified:-H0: The means of all driving conditions or traffic are equal.-H1: The mean of at least one driving condition or traffic group (control or smart, without or with cyclists) is different.

## 3. Results and Discussions

Statistical processing using ANOVA has produced some interesting results. In detail, by representing them using box-and-whiskers plot graphs and tables, it is possible to deduce appropriate considerations responding to the objectives of this research.

### 3.1. One-Way ANOVA with Dependent Variable Acc X

In Figure 5, the Acc X dependent variable has been analyzed. The four driving conditions reported in abscissa refer respectively to: (1) control condition, without cyclists; (2) control condition, with the presence of cyclists; (3) smart condition, without cyclists; (4) smart condition, with the presence of cyclists.

The outcome of the ANOVA, provided also in Table 1, shows that there are no statistically significant deviations between the four conditions.

The presence of cyclists in the same lane forces the driver to pass them, and this maneuver, evidently, is not performed by increasing the acceleration of the vehicle in a statistically significant way compared to the condition of an isolated vehicle. 

The presence of an OBU, however, could induce a modification of the user’s behavior and, consequently, have effects on the longitudinal acceleration. However, even this condition does not statistically differ from the other reference scenarios. The meaning seems clear: a winding road is the main constraint for the driver’s travel, and they do not alter their behavior even with further disturbances (cyclists) or support tools (OBU).

Lastly, it can be noted that the averages of the longitudinal acceleration in all four conditions are, however, low, thus confirming again what has been just stated.

The positive aspect of this result is that the OBU does not lead to an overload, even in conditions of greater stress, such as the scenario with the cyclists (4).

Concerning Table 1 (and similarly Table 2, Table 3, Table 4 and Table 5), the meanings of the titles in the first row are as follows:-Source: is the name of the variable;-SS is the sum of squares due to each source;-df is the degrees of freedom associated with each source;-MS is the mean squares for each source (that is the ratio SS/df);-F is the F statistic, that is the ratio of the mean squares;-Prob > F is the *p*-value, that is the probability that the F statistic can assume a value larger than the computed test statistic value.

### 3.2. One-Way ANOVA with Dependent Variable Acc Y

In Figure 6 and Table 2, the Acc Y dependent variable has been represented. The four driving conditions are the same of the previous case.

From a statistical perspective, there are no significant differences between the first three conditions, while the fourth shows a certain level of deviation. This condition is considered the most challenging for the user as it includes the presence of both cyclists and an OBU that could potentially lead to excessive engagement. However, it should be noted that this condition also differs from the others as the mean value is slightly lower. As is well known, the lateral acceleration is dependent on the square of the speed and is inversely proportional to the radius of the trajectory. In this case, it is likely that the presence of cyclists or a possible distraction caused by the OBU may have led to sharp steering movements, resulting in a decrease in the radius and an increase in the value of this variable. However, this did not occur as the user in this scenario adopted a more cautious behavior with trajectories characterized by wider radii.

Although the box plot in Figure 6 seems to show some differences in the averages of the first three conditions, their magnitude is too small to allow for any meaningful conclusions. The lines extending beyond the box represent the expected deviations as determined by the ANOVA analysis. It can be observed that the greatest dispersion of Acc Y occurs in the first driving condition, where the user is in control condition and there are no cyclists. This result suggests that the absence of external constraints, such as cyclists or violation-signaling devices, allows the user to exercise greater freedom in their driving behavior, which may vary according to their individual driving skills. However, it is worth noting that the sample size may not be sufficient to fully capture these variations. A larger sample size may have led to a more accurate representation of the data.

**Table 2 sensors-23-01758-t002:** ANOVA results for Acc Y.

Source	SS	df	MS	F	Prob > F
Acc Y	2.0234	3	0.67446	3.27	0.0283
Error	10.7189	52	0.20613		
Total	12.7423	55			

### 3.3. One-Way ANOVA with Dependent Variable Road Gap

Figure 7 and Table 3 show the road gap dependent variable. The four driving conditions are the same as the previous cases.

This measure indicates the distance of the center of the vehicle from the right edge of the road section. In the present case, the overall section is 7.00 m wide, 3.50 m of which is dedicated to a lane, plus a small shoulder for each side. It means that the vehicle is perfectly in the middle of its lane when the road gap is equal to 1.75 m. In this condition, as the vehicle is about 1.60 m wide, the distance between its left edge and the central axis of the road is 3.50 − (1.75 + 0.80) = 0.95 m. Figure 7 shows that the road gap variable took values of around 3.00 m in driving condition 4, and then the vehicle crossed the center line (separating the two opposite directions of traffic) for about (3.00 + 0.80) − 3.50 = 0.30 m. This result, on such a winding road and considering the presence of cyclists (and, therefore, a further deviation from the right edge during the passing), can be considered widely acceptable.

The analysis of Figure 7 shows no substantial differences between the four conditions, except for conditions 2 and 4. However, the averages of these two conditions were slightly higher. This is perfectly normal as cyclists travel in queues one behind the other along the right side of the roadway; when overtaking, the driver of the vehicle moves away at a distance of almost one meter from them to ensure a sufficient margin of safety. The presence of the OBU slightly increases the value of this variable but does not lead to statistically significant differences with the other two conditions.

**Table 3 sensors-23-01758-t003:** ANOVA results for the road gap.

Source	SS	df	MS	F	Prob > F
Road Gap	1.21928	3	0.40643	3.36	0.0257
Error	6.29671	52	0.12109		
Total	7.51599	55			

### 3.4. One-Way ANOVA with Dependent Variable Speed

Figure 8 and Table 4 present the dependent variable, speed. The four driving conditions, as previously noted, are the same.

The statistical analysis reveals significant differences in the speed between certain conditions, which, at a first glance, may appear surprising. It can be observed that the two conditions (2 and 4) with cyclists exhibit higher speeds. A reduction in speed would, instead, have been expected as cyclists could have represented an obstacle for the driver. 

However, as cyclists are positioned in a row along the right edge, they pose a risk of accident but do not constitute a significant obstruction to visibility or a significant challenge for passing. Overtaking occurs without significant longitudinal acceleration (as seen in Figure 5 and Table 1), albeit with a higher speed to minimize the duration of the maneuver and, therefore, reduce the likelihood of collision with cyclists. 

Finally, it can be observed that the presence of the OBU is not significant in the absence of cyclists (conditions 1 and 3) but leads to better results (in terms of average and lower dispersion). 

This indifference of the OBU should be interpreted positively as it does not cause disturbances compared to the condition in which it is absent.

**Table 4 sensors-23-01758-t004:** ANOVA results for the speed.

Source	SS	df	MS	F	Prob > F
Speed	2114.04	3	704.68	26.22	<0.0001
Error	1397.52	52	26.875		
Total	3511.56	55			

### 3.5. One-Way ANOVA with Dependent Variable Steer Speed

Figure 9 and Table 5 present the dependent variable and steer speed. The four driving conditions, as previously mentioned, are consistent throughout the analysis.

The statistical analysis reveals notable differences in this variable. Specifically, in the absence of cyclists (conditions 1 and 3), regardless of the presence of the OBU, the values of steer speed were significantly higher than in conditions with cyclists (conditions 2 and 4). This behavior can be attributed to the fact that in isolated conditions, the user perceives fewer constraints, leading to a greater focus on steering to minimize the risk of collision. As previously stated, the presence of the OBU does not significantly alter the results, but it does result in a lower dispersion between conditions 2 and 4. This suggests that the OBU does not have a negative impact on the results and instead promotes a more uniform driving behavior.

**Table 5 sensors-23-01758-t005:** ANOVA results for the steer speed.

Source	SS	df	MS	F	Prob > F
Steer Speed	5323.89	3	1774.63	45.93	<0.0001
Error	2009.01	52	38.63		
Total	7332.90	55			

### 3.6. One-Way ANOVA with Dependent Variable OW (Overall Worklod)

In this case, for the sake of brevity, no graphs and tables are shown. The analysis was carried out considering only two scenarios: absence and presence of the OBU. The results lead to a Prob > F of 0.681 (F value= 0.17), and, therefore, there is no statistically significant difference between these two conditions. The result is interesting as the users’ perception is perfectly in compliance with the instrumental results described by the other indicators. The OBU itself does not represent an additional load on the driver’s capacity. This demonstrates the utility of quantifying workload with measurements from different classes. It is opportune to remember that NASA TLX is determined through a questionnaire after the test and, although it takes into account many aspects of the task, it does not allow to record its value continuously during the driving.

### 3.7. Discussions

The in-depth analysis of the results makes it possible to summarize some general important considerations:-The classification of workload measurements into three main groups (subjective, performance, and physiological) should suggest that the experimenter should use at least two different types to gain a broader understanding of the observed phenomenon.-Even within the same class (performance measures, as in this research), a large number of variables should be used; otherwise, the user’s driving behavior remains unclear. In this research, for example, an examination of the variables relating to the longitudinal acceleration, the speed, and the steering speed permitted, in the best way possible, the interpretation of the passing maneuver of cyclists, often trivialized through unrealistic theoretical schemes.-The presence of the OBU has never caused difficulties while driving, but in some cases, it has improved the information acquisition. For further refinement, the authors could start their investigation from these results to modify the architecture of the OBU (in graphic terms, introducing sound messages, etc.) in order to further improve its impact in the conditions in which its presence is expected.

In conclusion, the indicators used in this research have frequently revealed statistically significant differences between the conditions with and without the presence of cyclists. This result is particularly evident for the trajectory (road gap), longitudinal speed (speed), and steering speed as the task of passing cyclists greatly influences these variables.

## 4. Conclusions

The research presented in this paper aimed to evaluate the effectiveness of certain workload measurements in a medium complexity driving environment, where the user was exposed to the use of an OBU. The study not only aimed to examine the contribution of the OBU in terms of any potential increase in workload, but also the effects of a complex scenario involving groups of cyclists to be passed. The results indicate that the response of these assistance systems is not uniform, but rather preferable in certain scenarios than others. This suggests the need for designing such instruments in a dynamic way to vary their contribution in compliance with the road conditions and the psycho-physiological characteristics of the user.

In terms of improving safety, modifying the architecture of the OBU may be a valid direction for improvement. However, the role of the road manager is also significant in this regard. The results of this study can provide useful information for the manager, such as identifying critical features of a particular route that can be mitigated in a rational way (e.g., additional signs, stricter speed limits, traffic calming devices, etc.).

As the digitization of roads continues to progress, it is important to establish a platform for continuous data exchange between stakeholders (infrastructure managers, driver aid manufacturers) through a web-GIS-based platform capable of processing and analyzing data to aid in decision-making. Future research will aim to evaluate additional scenarios considered highly critical and identify any additional synthetic indices for characterizing driving behavior.

## Figures and Tables

**Figure 1 sensors-23-01758-f001:**
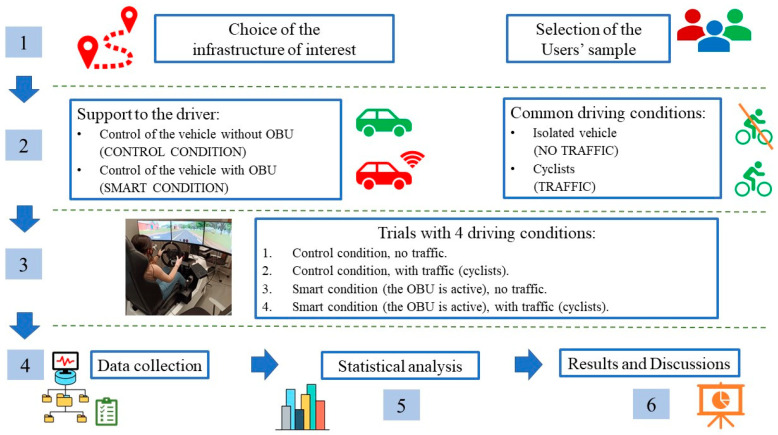
Flow-chart scheme of the proposed procedure.

**Figure 2 sensors-23-01758-f002:**
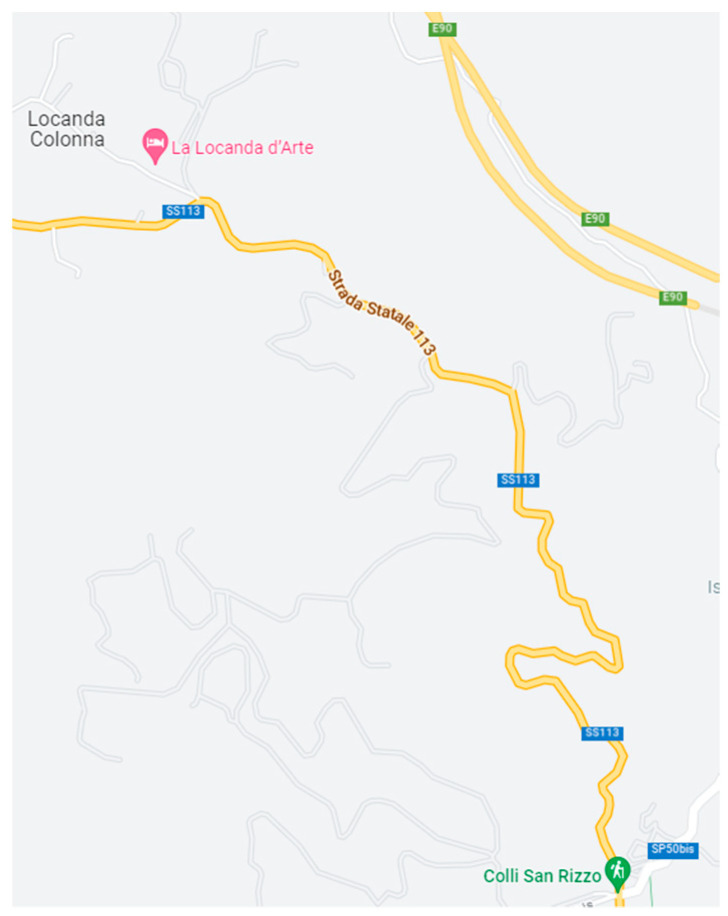
The rural road called SS113, reported in a simulated environment, connects the localities of “Colli San Rizzo” and “Locanda”.

**Figure 3 sensors-23-01758-f003:**
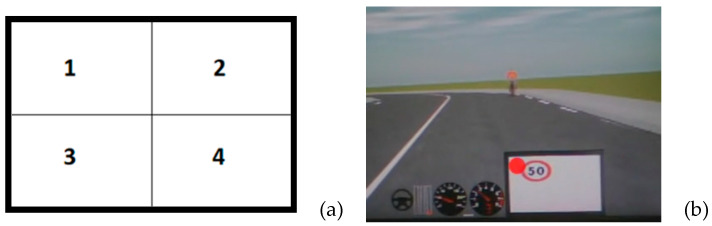
(**a**) Detail of the OBU screen with a subdivision of the display according to the type of warning. (**b**) Note the space dedicated to the OBU (a white rectangle in the center of the screen) in which there is a warning of exceeding the speed limit of 50 km/h.

**Figure 4 sensors-23-01758-f004:**
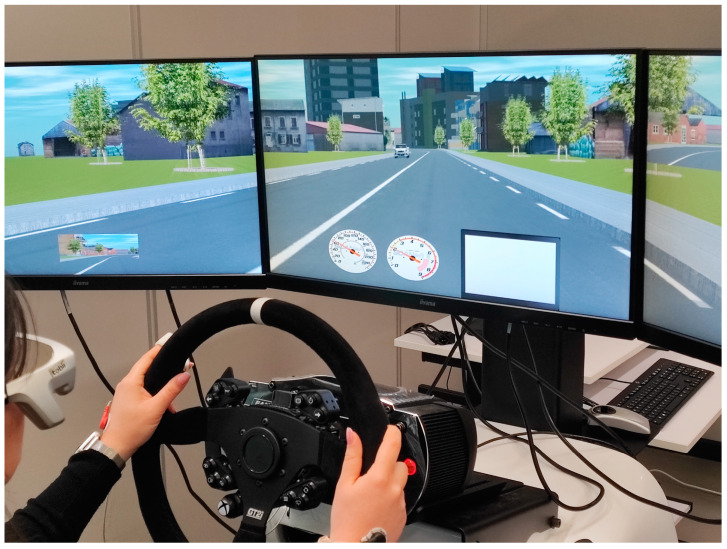
The driving simulator of the Laboratory of Road Infrastructure in Messina (Italy).

**Figure 5 sensors-23-01758-f005:**
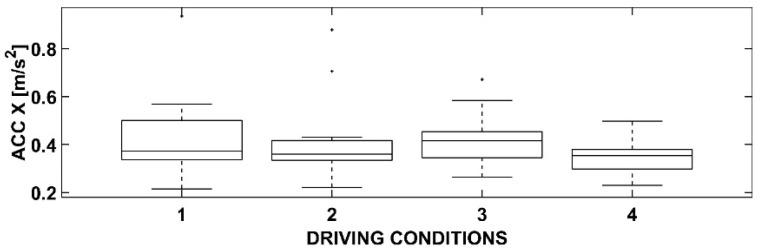
Box-and-whiskers plot for Acc X.

**Figure 6 sensors-23-01758-f006:**
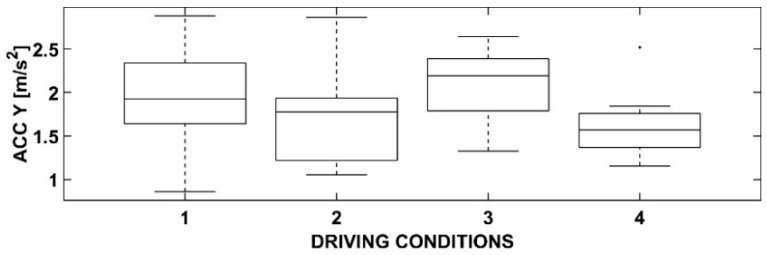
Box-and-whiskers plot for Acc Y.

**Figure 7 sensors-23-01758-f007:**
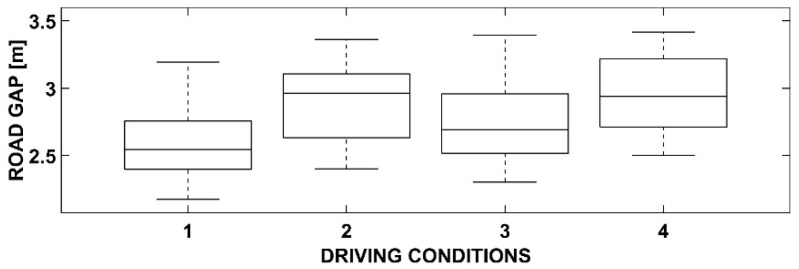
Box-and-whiskers plot for the road gap.

**Figure 8 sensors-23-01758-f008:**
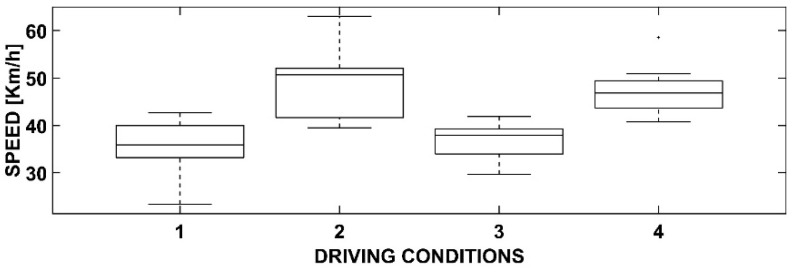
Box-and-whiskers plot for the speed.

**Figure 9 sensors-23-01758-f009:**
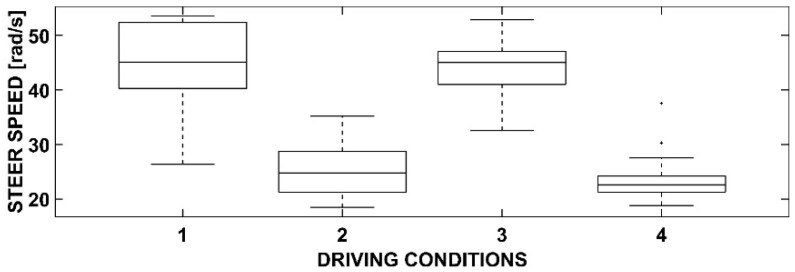
Box-and-whiskers plot for the steer speed.

**Table 1 sensors-23-01758-t001:** ANOVA results for Acc X.

Source	SS	df	MS	F	Prob > F
Acc X	0.06667	3	0.02222	1.13	0.3448
Error	1.02118	52	0.01964		
Total	1.08785	55			

## Data Availability

Some of the data used in this study may be available from the corresponding author upon rea-sonable request.

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
