# Peer review of "The Effects of ADAS on Driving Behavior: A Case Study"

_sensors, 2023, doi:10.3390/s23041758_

Round 1
Reviewer 1 Report
Introduction section is interesting but it seems to be out of the topic Especially point 1.1 Research gap seems to be slightly out of the previous context. I recommend to join studies of “Workload” of drivers with ADAC system more precisely to focus reader on the remaining scope of the studies.
Lines 289-293 – does smart conditions means that OBU is active? It is not clear for me.
Figure 6 – present description of points 1-4 below the figure.
Figure6 – in my opinion there is no effect of driving conditions on ACCX.
Figure 7 – how to explain differences in deviation bars ?
Section 3.3, Figure 8 and section 2.6 line 254 – Lane Gap or Road Gap – unify terms.
I recommend to summarize all of considered factors in term of ANOVA results. Which of factors (Acc x, Acc y, Lane Gap, Speed etc..) is statistically different, and which of them are not.
Author Response
Please, see the attached file

Reviewer 2 Report
Dear authors,
For the current development trend of intelligent and connected vehicles, the theme of this manuscript is interesting and challenging. The authors recruited 14 subjects to complete the driving experiment with a driving simulator and analyzed the obtained data using statistical methods, such as ANOVA. At first glance, the research is solid. However, as a work of experiments- data collection - statistical analysis, the value and contribution of this study depend on the quality of the data obtained. According to the results of previous research in the research field, the acceptance, utilization, and impacts of OBU represented by ADAS vary greatly among drivers with different characteristics (gender, age, culture, driving experience, driving tendency, and so on). Therefore, it is impossible to answer this question, the effects of ADAS on driving behavior, with the data of only 14 drivers with similar characteristics. It is suggested that the authors change the title of their work to narrow and focus the research theme. In other words, the authors' work and results are unacceptable with the current theme. Moreover, please standardize your academic writing, including removing unnecessary newlines and using figures and tables normally.
Hope my comments can help you improve your work. Wish you success.
Sincerely,
The reviewer
Author Response
Please, see the attached file

Reviewer 3 Report
The paper entitled "A Procedure to Verify the Effects of ADAS on Driving Behavior" proposes a methodology to measure how drivers' behavior is affected by ADAS components in vehicles.
Major remarks:
- Please provide a system-level architecture of the whole setup to make it easier to understand, and then continue with the whole information presented in each subsection from Section 2, which in the current version of the manuscript seem unlinked to each other.
- Since you are presenting statistical data, I think that data coming from just 14 users is too less to draw a concise and practical conclusion. Obviously, the drivers on the roads are not just students-alike, so you should definitely comment on that. How does this affect the obtained results?
- Aren't both the dashboard gauges and the rectangle too small illustrated on the display (as in Fig. 2)? Doesn't this already affect the user if it struggles to see what's represented/written in that rectangle?
- Where is the rectangle from Fig. 4 represented in Fig. 3?
- Please give more details about the information presented in Table 1; it is difficult to understand the meaning of the first line elements and of course their correspondence to the ones on the first column. The same for the other tables.
- What do the horizontal lines on the edges of Fig. 5 represent?!?
Author Response
Please, see the attached file

Round 2
Reviewer 2 Report
Dear authors,
It is an honor that my comments can truly help you. Glad to see the manuscript have been improved. To publish this manuscript, you should alter the title correctly to successfully narrow your research aim. For your reference, I think this, the effects of ADAS on driving behavior: a case study, may be a good one. On the other hand, the writing should be carefully proofread.
Wish you success!
Sincerely,
The reviewer
Author Response
please, see the attachment

Reviewer 3 Report
The authors should still answer my first comment: "Please provide a system-level architecture of the whole setup to make it easier to understand". It would be really useful to add a diagram (figure) with the architecture of the whole setup.
Author Response
please, see the attachment
